# Secondary or Specialized Metabolites, or Natural Products: A Case Study of Untargeted LC–QTOF Auto-MS/MS Analysis

**DOI:** 10.3390/cells11061025

**Published:** 2022-03-17

**Authors:** Franz Hadacek

**Affiliations:** Plant Biochemistry, Albrecht-von-Haller Institute for Plant Sciences, Georg-August University of Goettingen, 37073 Goettingen, Germany; franz.hadacek@biologie.uni-goettingen.de

**Keywords:** untargeted metabolomics, auto-MS/MS spectra, specialized metabolites, identification

## Abstract

The large structural diversity of specialized metabolites represents a substantial challenge in untargeted metabolomics. Modern LC–QTOF instruments can provide three- to four-digit numbers of auto-MS/MS spectra from sample sets. This case study utilizes twelve structurally closely related flavonol glycosides, characteristic specialized metabolites of plant tissues, some of them isomeric and isobaric, to illustrate the possibilities and limitations of their identification. This process requires specific software tools that perform peak picking and feature alignment after spectral deconvolution and facilitate molecular structure base searching with subsequent *in silico* fragmentation to obtain initial ideas about possible structures. The final assignment of a putative identification, so long as spectral databases are not complete enough, requires structure searches in a chemical reference database, such as SciFinder^n^, in attempts to obtain additional information about specific product ions of a metabolite candidate or check its feasibility. The highlighted problems in this process not only apply to specialized metabolites in plants but to those occurring in other organisms as well. This case study is aimed at providing guidelines for all researchers who obtain data from such analyses but are interested in deeper information than just Venn diagrams of the feature distribution in their sample groups.

## 1. Introduction

Secondary, or specialized, metabolites, according to more recent suggestions [1], comprise all those that are not important to sustain life but rather contribute to adaptation and survival. Primary or central metabolites, by contrast, represent all those that are indispensable to maintain life. Whereas their number is estimated to comprise around 10,000, specialized metabolites probably amount to several hundreds of thousands [1]. Other names for specialized metabolites are natural products or phytochemicals, terms that are used in pharmaceutical sciences to distinguish drugs of natural origin from synthetic ones. Specialized metabolites occur in all prokaryotic and eukaryotic organisms, with a tendency to become more scarce in those organism groups that developed an advanced immune system, such as vertebrates [2].

When the term ‘metabolomics’ was coined, the first published studies focused on the central metabolism. The predominantly used method was GC–MS, gas chromatography linked to a mass spectrometer with electron impact ionization (EI) [3,4]. The digitization of data acquisition facilitated accessing commercial and freely available spectra libraries. Among the latter, the Golm Metabolome Database (GMD) [5] and the FiehnLib [6] provide the most extensive mass spectral data collections. GC is especially suitable to separate sugars, but it also allows analyzing of amino and organic acids as well as many other central metabolites. Its major constraint is that it fails to separate larger molecules, for example, larger peptides and oligosaccharides, the upper limit being 600–650 Da. Volatility represents a major constraint. Very small terpenoids and simple aromatic specialized metabolites, all of them highly volatile and components of odors that are produced by living organisms, can be analyzed efficiently by GC–MS [7,8,9,10,11]. A large commercially available database exists, NIST [12], which is usually bundled with the instrument and contains a huge dataset of volatile specialized metabolites.

Specialized metabolites with higher molecular weights and with specific structural features, e.g., lactone rings, may not survive the high-temperature volatilization (250 °C or even higher) of the GC sample injection procedure. In these cases, LC–MS, liquid chromatography linked to mass spectrometry, remains the most commonly available method of choice. The fundamental difference of the EI spectra from the GC–MS analyses is that LC–MS analyses yield chemical ionization (CI) mass spectra that only show a few features, which represents a disadvantage for structure elucidation attempts. In this context, collision-induced dissociation experiments (CID) of specifically selected ions help to increase fragmentation (MS/MS), and many MS/MS library spectra have been generated by this technology using different collision energies [13,14]. 

Considering the often limited instrument time, auto-MS/MS techniques became more prominent, of which two types exist: data-dependent data acquisition (DDA, this paper), which provides information about the precursor ion, and all-ion fragmentation or SWATH mode (sequential window acquisition of all theoretical mass spectra), which provides more MS/MS spectra within a given time-window but supplies no precursor information [14]. The aim of this paper is to provide a case study on specialized metabolite identification attempts on the basis of auto-MS/MS spectra that were obtained in the data-dependent acquisition mode. 

Recently published reviews, e.g., Refs. [14,15], offer a good overview of which software and databases are available to perform structure elucidation on the basis of MS/MS data. In addition, an extensive survey of the available databases for compound identification of specialized metabolites exists [16]. Important criteria for the choice of the software, at least in the opinion of the author, are free availability, efficient data spectra deconvolution properties, and usability on a local computer. Free availability is especially important in the case of non-instrument owners who are then able to analyze data without having to find access to expensive licensed software. Peak picking and alignment have to be done by the software due to the huge data amounts, and deconvolution of co- or closely eluting analytes has to deliver spectra that are suitable for library searches. Web-based software often requires lengthy upload and analysis times of your data and provides less manual exploration and adjustment possibilities for the obtained results. The author has good experiences with the software MS-DIAL [17,18,19], but many other suitable software tools exist [20]. 

Tandem mass spectral data, MS/MS, are required for structure elucidation. Identification on the basis of accurate masses alone is too ambiguous due to the existence of high numbers of isomeric and isobaric metabolite structures, many of which may even belong to different compound classes. MS/MS spectral data are especially expected in the case of reporting novel structures [21] when preparative isolation and structure elucidation on the basis of a combination of UV, IR, MS, and NMR techniques [22] are impossible. Spectral databases are incomplete as they are mostly limited to spectra from commercially available standards. In attempts to overcome this constraint, molecular structure databases have been recommended as a primary search tool for unknowns [23]. In order to do this, special software with *in silico* fragmentation capabilities is required, for example, MS-FINDER [18,24], or CSI:FingerID [25], which is included in the SIRIUS software framework [26,27,28], both of which are available freely, but various other similar software tools are also available [20]. Despite existing comprehensive guidelines for metabolomic studies [21,29,30], only the first published [21] mentions SciFinder^n^ from Chemical Abstract Service (CAS) [31] and none of them Reaxys from Elsevier [32], two of the largest existing databases for chemicals, including specialized metabolites. Especially in assumptions that the majority of analytes represent ‘known unknowns’, the use of CAS was recommended [33,34].

Flavonoids represent a class of specialized metabolites that are very common in plant tissues [35], and, for this reason, they were chosen for this case study. Specifically, flavonol trisaccharides with a mass of 756 and 772 Da will be focused on to illustrate the identification problems of isomeric metabolites. In three of the exemplary flavonoid glycosides, a phenylpropanoid acid replaces one sugar, which only in two cases increases the mass to 816 Da. The MS/MS spectra of the exemplary flavonol glycosides derive from an ongoing study that focuses on physiological traits that associate with the development of apomixis (asexual propagation) in polyploid species of goldilocks (*Ranunculus auricomus* L.), a herbal species that can become very common in less extensively used meadow habitats, especially in Europe and adjacent Asia [36].

This case study aims to provide a guideline for all those who want to start with auto-MS/MS-based identification of specialized metabolites. In this context, different issues will be explored: (1) suitability of auto-MS/MS spectra for identification; (2) limiting structural candidates with the help of molecular structure databases and *in silico* fragmentation; and (3) SciFinder^n^ as a tool for finding information about structural candidates, which, ultimately, is also necessary.

## 2. Materials and Methods

### 2.1. Plant Material

The origin of the plant material, buds of *Ranunculus auricomus*, is described elsewhere [36]. Bud material was stored frozen at –20 °C.

### 2.2. Chemicals

Methanol (MeOH) was LC–MS grade (Th. Geyer, Hoexter, Germany); ethyl acetate (Etac) and formic acid p.a. (Honeywell Fluka, Bucharest, Romania) and water purified by an Arium Pro system (Sartorius, Göttingen, Germany).

### 2.3. Extraction and Fractionation

Several buds, 5–8(10), from one individual were freeze-dried (ZM200, Retsch, Haan, Germany) and extracted with 1 mL MeOH. After drying on a SpeedVac (RVC 2–25, Martin Christ Gefriertrocknungsanlagen, Osterode am Harz, Germany), the residue was dissolved in 0.7 mL water. To achieve optimal dissolution, the samples were incubated in dark for 48 h. Then, 0.7 mL ethyl acetate (p.a.) was added and the sample thoroughly mixed. For efficient phase separation, all samples were incubated for 24 h in dark. Glass vials were standard autosampler vials (Macherey Nagel, Düren, Germany). The MeOH and Etac phases were dried separately on the SpeedVac and stored in argon atmosphere at −20 °C until further processing. The yields of the Etac and water phase varied, 0.1–2.9 mg for the former and 0.7–12.8 mg for the latter.

### 2.4. Sample Preparation

The Etac phase was dissolved in MeOH:water:formic acid (25:75:0.1, *v*/*v*/*v*) to yield a final concentration of 0.1 mg/mL. Samples were injected without further filtration.

### 2.5. LC–TOFMS Analysis

Samples were analyzed by an Agilent 1290 Infinity II LC system coupled to an Agilent G6545A quadrupole time-of-flight mass spectrometer (QTOF, Agilent Technologies, Waldbronn, Germany). The LC system consisted of a G7167A multisampler, a G7104A quaternary pump, and a G7117B DAD UV detector (Agilent Technologies, Waldbronn, Germany). The column was an Agilent Zorbax Eclipse C18 column, 1.8 µm particle size, 100 × 2.1 mm (Agilent Technologies, Waldbronn, Germany). The MS interface was an orthogonal electrospray ionization source (ESI).

The solvent gradient started from 95% A (water + 0.1% formic acid) and changed to 100% B (MeOH + 0.1%) in 15 min that was kept for further 5 min. The flow rate was 0.4 mL/min. The column compartment temperature was set to 35 °C. The DAD recorded spectra from 220 to 580 nm. The injection volume was 8 µL.

All samples were analyzed in the positive and negative ion-mode separately with otherwise identical parameters. The acquisition range was set from *m*/*z* = 100–1700 with a scan rate of 2 spectra/sec. The ESI parameters were set as follows: sheath gas flow was 11 L/min, the sheath gas temperature 350 °C, the nebulizer pressure 35 psig, the gas flow 8 L/min, the scan source parameters VCap 3500 V, nozzle voltage 1000 V, fragmentor 175 V, and skimmer1 65 V. Parameters for the auto-MS/MS mode were as follows: *m*/*z* = 20–1000, scan rate 3 spectra/sec. The collision energy was set according to mass: 100 (10 eV), 300 (26 eV), 500 (45 eV), 700 (60 eV). The precursor selection was defined as follows: max precursor per cycle 1, threshold abs. 1, threshold (rel., %) 0.01, purity stringency (%) 100, purity cutoff (%) 30, isotope model common, active exclusion enabled but released after 2 spectra and after 0.5 min, precursors sorted after abundance only, and isolation width narrow (~1.3 Da). In the positive mode, reference masses were 121.0508773 and 922.009798, and, in the negative mode, 119.03632 and 966.000725. The choice of the parameters followed recommendations of the Agilent technician with a slight modification (max precursor per cycle) [37].

### 2.6. Data Analysis

The instrument software was MassHunter 8.0.0 (Agilent Technologies, Waldbronn, Germany). Data files were converted by AnalysisBaseFile Converter [38] for import into MS-DIAL 4.x [17]. Peak picking, alignment, deconvolution, and first MS/MS database search was performed with the following parameters: MS^1^ tolerance 0.01 Da, MS^2^ tolerance 0.05 Da, minimum peak height 3000, mass slice width 0.1 Da, sigma window value 0.5, retention time tolerance 0.05 min, MS/MS databases Vaniya-Fiehn Natural Product Library, BMDPS-NP, GNPS, MassBank, and MetaboBASE from the software’s website. Molecular structure searches were performed with MS-Finder 3.52 [24] and Sirius 4.9 [26,27,28]. Chemical structures, InChI keys, and canonical SMILES (Simplified Molecular Input Line Entry Specification) were created with ChemDraw 20.1.1. (PerkinElmer Informatics, Rodgau, Germany).

### 2.7. Identification and Nomenclature

Pure trivial names of putatively identified flavonoid glycoside structures are avoided except for well-known flavonoid ring structures (isorhamnetin, luteolin, kaempferol, and quercetin). Generally, it is more or less impossible to differentiate between isomeric hexoses (glucose and galactose) and pentoses (xylose, arabinose, and apiose). If [M + Na]^+^ MS/MS spectra are available, glucose and galactose and even different linkage types between them and rhamnose can be differentiated [39]. In all other cases, hexoses are assumed to be glucose, deoxyhexoses to be rhamnose, and pentoses to be xylose. Sugars are abbreviated (gal, galactose; glu, glucose; rha, rhamnose; xyl, xylose). Bonds between sugars are not specifically indicated if they are (1→6) (glucose) or (1→4) (rhamnose). Positions on the first sugar linked to the flavonoid are indicated by “ and the second sugar by ”.

Substitution patterns on the flavonoid ring system were elucidated based on the prevalence of [Y_0_ − H]^●−^ (3-subst.), [Y_0_]^−^ (7-subst), and [Y_0_ − 2H]^−^ (3,7-disubst.) [40]. MS/MS fragmentation of flavonoid rings were compared to full collision energy ramp MS/MS spectra [41]. Trisaccharide fragmentation was compared to available CID studies [39,42,43,44].

## 3. Results

Figure 1 shows the [M − H]^−^ spectra from the twelve selected flavonol glycosides. The auto-MS/MS method generates spectra in which the product ions of the flavonoid ring system are very conspicuous and hint the substitution pattern of the ring system. The number of product ions is much higher for the glycoside part of the precursor ion and they are thus less clearly visible and more difficult to interpret. Table 1 lists the identified product ions that contributed to identification of the 12 flavonol glycosides. All were detected in the negative mode, and only four of them in the positive mode. In all cases, both [M + H]^+^ and [M + Na]^+^ precursor ions were detectable. Only one sodium adduct failed to yield an MS/MS spectrum.

Sugar isomers cannot be differentiated on the basis of MS/MS spectral data, and reporting them as canonical SMILES does justice to this deficiency. Glucose, rhamnose, and xylose represent the most commonly reported sugars in flavonol triglycosides [45], and thus a hexose was assumed to be a glucose, a deoxyhexose a rhamnose, and a pentose a xylose in those cases in which no information from [M + Na]^+^ MS/MS spectra were available. The main reason for this approach was to facilitate database indexing of all compounds.

Appendix A contains all the important information for database indexing, an overview of the structures, trivial names used in the text, semisynthetic IUPAC names, identification level, SMILES, and the most closely resembling CAS registry numbers. In the figures, tables, and the text, a modified trivial nomenclature is used that aims to keep the names as short as possible without losing structure information (see Section 2.7).

### 3.1. Quercetin-3-glu-rha-7-glu (6.20 min)

The prominent [Y_0_─2H]^−^ feature at 299 Da pointed to a 3,7-disubstituted quercetin derivative [40]. The MS/MS spectral search by MS-DIAL only yielded a single hit, isorhamnetin-3-rha-rha-rha. MS-FINDER proposed quercetin-3-glu-rha-7-gal; SIRIUS provided no structural suggestions. The only detectable glycoside feature of 463 Da pointed to quercetin-3-glu or quercetin-7-glu. The 271 and 151 Da fragment ions have also been detected in quercetin-3-glu [41]. The finally assigned structure, quercetin-3-glu-rha-7-glu, belongs to the more commonly occurring flavonol glycosides [46,47,48,49,50].

### 3.2. Quercetin-7-rha-(4″-glu)-glu (6.78 min)

The prominent [Y_0_]^−^ feature at 301 Da hinted a 7-substituted quercetin [40]. The MS/MS spectral search by MS-DIAL found the same spectrum as for the flavonol glycoside at 6.20 min. MS-FINDER structural proposals included a range of 3- and 3,7-disubstituted quercetin derivatives. SIRIUS’s, by contrast, provided no flavonoid structure proposals at all.

Features at 283 Da—this one especially—and 271 Da, and 151 Da provided support for the first sugar being a rhamnose [41]. The feature at 446 Da provided further evidence that sugars 2 and 3 are both glucoses. The two remaining glucose moieties may connect 1→4, in a similar way as it was shown for synthetic flavonol glycosides [51]. Neither the finally assigned structure nor the possible isomer quercetin-7-(2″-glu)-rha-glu are reported in the literature.

### 3.3. Kaempferol-3-glu-rha-7-glu (7.35 min)

The prominent [Y_0_]^−^ feature at 285 Da pointed to a 7-substituted kaempferol, but the additional presence of a [Y_0_ − 2H]^−^ feature at 283 Da indicated a 3,7-disubstituted kaempferol [40]. MS-FINDER proposed kaempferol-3-glu-glu-7-rha, similar to the MS/MS library search in MS-DIAL, and SIRIUS’s kaempferol-3-glu-rha-glu.

The structure of kaempferol-3-glu-rha-7-glu was supported by the observed in-source fragmentation. A feature at 593 Da indicated the loss of a glucose, and a feature at 447 Da the further loss of a rhamnose. The features 357 and 241 Da characterize a kaempferol-3-glu structure [41]. The proposed presents one of the more commonly occurring flavonol glycosides [46,49,52,53].

### 3.4. Kaempferol-3-(2″-glu)-glu-rha (7.46 min)

The prominent [Y_0_ − H]^●−^ feature at 284 Da indicated a 3-substituted kaempferol [40]. The MS/MS spectral search by MS-DIAL found close similarity to kaempferol-3-glu-rha-glu. MS-FINDER proposed 4,7-disubstitued kaempferol glycosides but also kaempferol-3-glu-rha-glu. SIRIUS suggested kaempferol-3-(2″-glu)-glu-rha.

Despite reports of 255 and 241 features [41], the observed 256 and 241 Da features can be regarded as congruent with a kaempferol-glu-rha feature. The additional glucose linked to the first glucose sugar could have caused this change. Consequently, the proposal from SIRIUS might be correct, but an isomer in the sugar part of the molecule is also possible. Kaempferol-3-(2″-glu)-glu-rha is mentioned in several studies [49,52,54].

### 3.5. Quercetin-3-glu-rha-7-rha (7.67 min)

In the negative mode, the prominent [Y_0_ − 2H]^−^ feature at 299 Da proposed a 3,7-disubstituted quercetin [40]. Based on this information, the MS/MS spectral search by MS- DIAL yielded quercetin-3-(2″-rha)-glu-rha, although the MS/MS spectrum of the MetaboBASE entry showed a prominent [Y_0_ − H]^●^^−^ feature and only a trace of the [Y_0_ − 2H]^−^. The corresponding MS_FINDER hit was quercetin-3-glu-rha-7-rha; SIRIUS, by contrast, failed to propose any flavonoid structure.

Features at 255 and 271 Da concurred with assumptions of a quercetin-glu structure, and a feature at 283 Da with a quercetin-rha structure [41]. A loss of glucose and rhamnose causes a 446 Da feature in the negative spectrum. This flavonoid glycoside was also detected as [M + H]^+^ and [M + Na]^+^. The [M + Na]^+^ MS/MS spectrum showed characteristic fragments for an rhaα6glu linkage [39]. The latter also showed a glu-rha feature at 347 Da (glu-rha). Quercetin-3-glu-rha-7-rha represents an often reported flavonol glycoside [55,56,57]. 

### 3.6. Quercetin-7-(2″-glu)-glu-rha (7.93 min)

In the negative mode, the prominent [Y_0_]^−^ feature at 301 Da indicated a 7-substituted quercetin [40]. The MS/MS spectral search by MS-DIAL only yielded a single hit, an isorhamnetin derivative. MS-FINDER suggested a 7-substituted kaempferol and luteolin glycosides; SIRIUS by contrast, no 7-substituted flavonol glycosides at all.

In this case, fortunately, several in-source features were visible in the positive mode, and the [M + H]^+^ adducts at 611, 465, and 303 Da proposed a quercetin-7-glu-glu-rha structure. Rather prominent features at 255 and 271 Da provided support for a quercetin-glu structure [41]. The MS/MS spectrum of the in-source [M + H]^+^ adduct at 303 Da showed high similarity to an MS/MS spectrum of quercetin deposited in Massbank (MB: PR309259). Tricetin, an isomer, was not found (HMDB0029620, [61]). The [M + Na]^+^ MS/MS spectrum contained fragments supporting a (2″-glu)-glu-rha trisaccharide structure [39]. The proposed structure is thus quercetin-7-(2″-glu)-glu-rha, which is not described in the literature.

### 3.7. Quercetin-3-(2″-rha)-glu-rha (8.04 min)

The prominent [Y_0_ − H]^●^^−^ feature at 300 Da indicated a 3-substitued quercetin derivative [40]. The MS/MS spectral search in MS-DIAL resulted in quercetin-3-glu-rha-rha. MS-FINDER proposed quercetin-3-(2″-rha)-glu-rha, and SIRIUS several isomers in terms of the sugar moiety.

The remaining question concerned the sugar linkage. In this context, the presence of a feature at 489 da pointed at quercetin-3-(2″-rha)-glu-rha [56], which also represents the most commonly reported one in the literature [58,59,60].

### 3.8. Luteolin-7-(2″-glu)-glu-rha (8.56 min)

The negative MS/MS spectrum showed a prominent Y_0_ feature at 285 Da that first suggested a 7-substituted kaempferol [40]. The MS/MS spectral search in MS-DIAL afforded kaempferol-3-glu-rha-(3″-rha). MS-Finder proposed 3,7-disubstituted kaempferol derivatives, SIRIUS the same structure as the MS-DIAL search. The flavonoid glycoside was also detectable in the positive mode and the in-source fragmentation product ion at 287 Da, which originally was thought to be kaempferol. The MS-DIAL search, however, revealed the spectrum to show the closest similarity to luteolin (BMDMS-NP 29252). The absence of features at 117 and 135 Da pointed to a linkage of the glycoside moiety to carbon 7 instead of 4′ [42].

The sugar moiety was assumed to be (2″-glu)-glu-rha, in analogy to quercetin-7-(2″-glu)-glu-rha, for which even a literature report exists on the basis of NMR data [62]. The [M + Na]^+^ MS/MS spectrum contained fragment ions that are characteristic of a 6α linkage between rhamnose and glucose and a 2β linkage between two glucoses. Fragments that characterize galactose were missing [39]. No reports on this flavonoid glycoside exist in the literature.

### 3.9. Isorhamnetin-3-(2″-glu-6″-feruloyl)-glu (8.70 min)

The prominent [Y_0_ − H]^●^^−^ and [Y_0_–CH_2_–H]^●^^−^ features, 300 and 314 Da, indicated a 3-substituted isorhamnetin derivative [40]. The MS/MS spectral search of MS-DIAL found no corresponding hits. MS-FINDER proposed various kaempferol and quercetin glycosides to which ferulic and sinapic acid is linked. SIRIUS failed to suggest any flavonoid structures.

A feature at 357 Da provided some evidence for a glucose on position three [41]. There exist now three possibilities how the additional glucose and the ferulic acid are linked to the glucose. One possibility is isorhamnetin-3-glu-glu-ferulic acid [63] and the other isorhamnetin-3-(2″-glu-ferulic acid)-glu [64], both of which were elucidated by 2D-NMR techniques. However, in consideration of a simple way to convert this ferulic acid–isorhamnetin glycoside to the structure of the one eluting at 9.50 min, isorhamnetin-3-glu-ferulic acid-7-glu, the structure of isorhamnetin-3-(2″-glu)-glu-ferulic acid is proposed. A similar linkage pattern was found in quercetin and kaempferol glycosides that form esters with coumaric, caffeic, and sinapic acid, the formation of which seems to be catalyzed by a specific flavonol-phenylacyltransferase [60]. All of these proposals pin the phenolic acid to glucose 6″ position. A feature at 175 Da adds support for a ferulic acid moiety.

The spectrum showed a product ion at 524 Da, for which, despite a high probability of deriving from the flavonol glycoside parent ion, no structure could be assigned to.

### 3.10. Isorhamnetin-3-(2″-xyl)-glu-7-rha (8.72 min)

A prominent [Y_0_]^−^ feature at 315 Da suggested a 7-substitued isorhamnetin derivative, but the presence of a [Y_0_–CH_2_–2H]^−^ feature at 299 Da pointed to a 3,7-disubstituted isorhamnetin derivative. The MS/MS spectral search by MS-DIAL failed to produce a hit. MS-FINDER proposed 3-substituted isorhamnetin derivatives, SIRIUS 3-substituted kaempferol and quercetin derivatives.

Similar to the previously described isorhamnetin derivative, a feature at 357 Da provided some evidence for a glucose on position 3 [41]. The sugar linkages follow a published structure [55].

### 3.11. Quercetin-3-(2″-p-coumaroyl)-glu-rha (9.11 min)

The prominent [Y_0_ − H]^●−^ feature at 300 Da indicated a 3-substitued quercetin derivative [40]. The MS/MS spectral search by MS-DIAL yielded quercetin-3-rha-glu-*p*-coumaric acid; MS-FINDER suggested the identical structure, and SIRIUS kaempferol-3-glu-(2‴-sinapic a.)-rha.

In-source fragmentation led to a feature of 609 Da that limited the linkage of the *p*-coumaroyl moiety to the glucose linked directly to quercetin. A feature of 395 Da even pinned the position of the *p*-coumaric acid to carbon 2″ on the glucose ring (for product ion structures, see Appendix A). Consequently, the proposed structure is quercetin-3-(2″-*p*-coumaric a.)-glu-rha [59].

### 3.12. Isorhamnetin-3-(6″-feruloyl)-glu-7-glu (9.50 min)

The prominent [Y_0_–CH_2_–2H]^−^ feature at 299 Da indicated a 3,7-disubstitued isorhamnetin derivative [40]. The MS/MS spectral search of MS-DIAL came up with hits for naphthoquinones, SIRIUS with a quercetin–sinapic acid glycoside.

A feature at 175 Da indicated a presence of ferulic acid, similar to isorhamnetin-3-(2″-glu-6″-feruloyl)-glu. The same mass pointed to an isomeric structure, for which isorhamnetin-3-(6″-feruloyl)-glu-7-glu is the most likely candidate. A series of similar kaempferol and quercetin glycosides in which sinapic acid serves as phenylpropanoid acid occurs in Arabidopsis [60].

The spectrum showed a product ion at 544 Da, for which, despite a high probability of being a derivative from the flavonol glycoside parent ion, no putative structure could be assigned.

### 3.13. UV-Spectra

Despite UV-DAD spectra being acquired in all the analyses, the analyte concentrations turned out to be too low to obtain interpretable spectra.

## 4. Discussion

### 4.1. Structure Identification and Feature Sorting on Basis of Auto-MS/MS Spectra

The exact masses that are obtained in the MS^1^ level do not suffice for structure elucidation. More complex specialized metabolites, such as flavonol glycosides, but also other specialized metabolite classes, e.g., saponins or steroid glycosides, require MS/MS spectra for more or less reliable putative identification. The loss or addition of a hydroxyl or a methyl group can happen both on the aglycone and the glycoside moiety, which results in a series of isomeric derivatives. Depending on which collision energies or energy ramps are used in CID or auto-MS/MS methods, the obtained spectra look different. Lower collision energies generate spectra with prominent precursor ions; higher collision energies cause a shift to product ions in the lower mass range [14]. Concerning flavonol glycosides, at low collision energies, you can study the glycoside structure, and, at higher collision energies, the flavonoid structure. The latter is the case for auto-MS/MS spectra, in which substitution patterns are more conspicuously visible than the sugar linkages. (Figure 1). In this study, a multistage energy ramp was used, depending on the molecular weight of the MS^1^ ion, to generate a pseudo MS^3^ spectrum that corresponds to an MS^3^ or MS^4^ CID spectrum [65]. Figure 2 illustrates the spectra that were obtained for luteolin-7-(2″-glu)-glu-rha (8.58 min). For this specific flavonol glycoside, it was possible to obtain a positive and negative MS/MS spectrum of the 755 Da ion and, in the positive mode, an MS/MS spectrum of the in-source fragment ion at 287 Da.

The retrieved library spectra for luteolin-7-(2″-glu)-glu-rha document that comparable spectra exist in MS/MS spectra databases (Figure 2). They are, however, much less represented in them than CID spectra and rarely sufficient for identification. CID spectra are acquired at much lower collision energies and show product ion fragments of the gradual fragmentation of the glycoside part of the flavonol glycoside. Such fragments were detected in some of the spectra of other flavonol glycosides in this study, but not in all. In this context, data from full collision energy ramp-MS^2^ spectra can prove as helpful [41].

In two cases, kaempferol-3-glu-rha-7-glu (7.35 min) and quercetin-7-(2″-glu)-glu-rha (7.93 min), in-source fragmentation occurred that provided more specific information about the glycoside structure, for the former in the negative and the latter in the positive mode. Consequently, due to their more or less accidental appearing occurrence, it can only be speculated which factors contribute to these phenomena, but they have to always be taken into account. In-source fragmentation is recognized as a useful phenomenon in

auto-MS/MS spectral analysis [66,67,68], and one study even explored it in context with flavonoids [69]. The authors proposed fragmentor voltages that were even higher than those used in this study, 230 and 330 V. At this point of the discussion, we must remind ourselves that complex mixtures of specialized metabolites, which are analyzed in untargeted metabolomics, contain different compound classes, and the pre-chosen analyses conditions will only be optimal for a few of them. To detect in-source fragmentation, the application of isotope labeling was recommended [68], but the focus of that study was on central metabolites. In the case of specialized metabolites, this approach would probably not be feasible. The structural diversity is much higher. In the case of luteolin-7-(2″-glu)-glu-rha, the in-source fragmentation leading to the detection of the flavonoid moiety product ion was fortunate because the MS/MS spectrum allowed the identification of luteolin, contrary to previous assumptions of kaempferol, an isomer to luteolin. In the present data, the existence of pronouncedly different sample groups, in which different flavonol glycosides occur, helped to recognize in-source fragmentation in the ion table (Table 2). This approach may not be methodically elegant but turned out to be practically efficient. The aligned ion table of MS-DIAL provides bar charts that inform about the signal intensities in different experimental groups. Similar bar chart patterns that point to related features are highlighted in red. These patterns, together with similarity in other parameters, such as signal–noise ratio, fold change, or *p*-values from an ANOVA of the sample groups, can provide hints to related adducts or in-source fragments, the feasibility of which still requires checking on the basis of a tentative structure. Depending on the filter values, an alignment analysis can yield 2000–20,000 features, and the challenge is to sort those that are more prominent in terms of amounts and offering MS/MS fragmentation for their identification. In-source fragments, however, may lack MS/MS spectra but still provide important information for structure identification. Recently, additional procedures have been proposed to clean feature lists, for example, the R-based tool MS-CleanR [70]. The latter could not be applied to the present dataset because of missing quality control analyses. The most recent guideline for untargeted LC–MS/MS analyses mentions quality controls, but not in context with feature filtering [29].

### 4.2. UV-Spectra

Along with MS/MS spectra, some studies also represent UV-spectra that can be obtained by the same analysis. There exist two inherent problems. One is that LC–MS/MS analyses require much lower concentrations than are suitable for LC–DAD analyses. This problem may become especially apparent with newer instrumentation. An efficient solution can be a split after the DAD in the ratio of 1:8 [71].

Another problem is solvent gradient times. In order to benefit from the information of UV spectra, they have to be pure. Often, slight shifts in the maxima or the appearance and disappearance of shoulders are highly indicative. This can be achieved with longer analyses times, but analyses times are often kept short by operators to facilitate higher sample throughput. In this study, the minimum elution difference of two flavonol glycosides is 0.02 min and lower. The alignment procedure makes it difficult to report exact values. A DAD would have failed to detect the co-eluting analytes even in the case of sufficient analyte concentrations. However, not only UV spectroscopy benefits from longer analyses times and the resulting improved chromatographic resolution; auto-MS/MS spectrometry would also yield better quality analysis results due to the fact that a lower number of chemical reactions will occur at the same time in the ion source.

### 4.3. Limiting Candidate Structures with Molecular Structure Databases

The previous section discusses some important aspects of MS/MS spectra that merit consideration during the identification process. MS/MS spectral libraries are not yet complete enough for providing precise automatized identification of MS/MS spectra, but hopes exist that the situation will improve in the future [23]. The use of structural databases in attempts to ameliorate this situation has its merits for certain. Even if the provided structures are not congruent or correct, the application of this software provides starting ideas on which structures are possible (Table 3). Only in one case, kaempferol-3-(2″-glu)-glu-rha, the CSI:FingerID module in SIRIUS listed the finally assigned structure as the first hit. In the other cases, MS-FINDER often came closer in terms of list ranking but never did the hit with the highest match factor concur with the finally proposed structure. Furthermore, MS-FINDER presents fragmentation ion products as chemical structures. One has to keep in mind that, in most cases, the shown structure represents only one of many possible isomers. Moreover, it certainly pays off to use more than one software tool. The implemented *in silico* fragmentation algorithms differ: MS-FINDER uses a rule-based method, CSI:FingerID in SIRIUS a combination of fragmentation trees and machine learning [20]. According to best-reporting practices [29], only kaempferol-3-(2″-glu)-glu-rha earns a C(ii) identification level and all others only C(iii) (Appendix A).

In addition to a survey about potential structures, ‘*in silico* fragmentation’ methods [23] can provide fragmentation tables of corresponding hits to the MS/MS spectrum in question, which proved extremely helpful to identify product ions that contribute to assign a putative structure for the analyte. Figure 3 illustrates the user interface of both software tools.

### 4.4. Searching the Literature and Making Yourself Searchable

The most frequent question that arises during the identification process of MS/MS spectra is if there exist papers that report MS/MS product ions of the candidate structure. This requires specific structure searches capabilities, and the most widely-used and comprehensive databases are SciFinder^n^ [31], CAS Registry via STN [72], and Reaxys [32].

SciFinder^n^ and Reaxys allow structure input by SMILES that can be generated by many different chemical structure drawing programs or the web tool PubChem Sketcher [76]. The advantage of SciFinder^n^ is that is represents both a reference (Chemical Abstracts) and substance database (CAS registry), both of which are linked. By contrast, Reaxys is more of a substance database that was developed from the discontinued Beilstein structure database [77].

When you start to work with SciFinder^n^, you will find out that the findings of quite a number of metabolomic studies are registered incompletely; i.e., the list of substances is incomplete or sometimes missing totally. The reasons for this deficiency are manifold: (1) often, metabolite lists are only provided as supplementary files and often hidden in spreadsheet files with several data sheets or in multipage PDF files; (2) the metabolites are designated by ambiguous trivial names on basis of which it is very difficult to conclude an unambiguous structure; and (3) if the glycoside moiety is only indicated as a simple sequence of hexose, deoxyhexoses, or pentoses, the structure information is regarded as too ambiguous for registration. Figure 4 summarizes the nomenclatural dilemma of specialized metabolites and points out which type names should be avoided for registration. In former times, IUPAC names had to be used. Though their generation can be achieved by software tools, they are too long to be practical.

As a result, structure searches do retrieve much fewer hits as would be possible in theory. Many of the retrieved references comprise studies that report the classical combination of spectral information, UV, IR, MS and NMR ^1^H, ^13^C, and various 2D experiments [22]. In recent years, however, they have become scarcer. This development may lead to a gap between the traditional exploratory research in organismic metabolite diversity and MS/MS-based metabolomics of central and specialized metabolites. To improve structure elucidation accuracy, LC–MS/MS can be combined with LC–NMR [15], but such state-of-the-art instrumentation is only available to a very small fraction of the scientific community that is engaged in untargeted plant metabolomics and research involving specialized metabolites in general [78].

In this study, none of the proposed structures for the twelve flavonol glycosides (Figure 1) could have been developed without consulting SciFinder^n^. What can be done to avoid incomplete registration of analysis results? A clear indication in the Results section to the existence of a table in the Appendix A that tabulates the detected metabolites should suffice [79]. This table, named Appendix A, should contain an overview of the identified structures with SMILES. They are the best identifier codes because InChI keys do not work for novel structures. Canonical SMILES are sufficient because MS/MS data provide no stereochemical information. Caution is always important in the assignment of structures to analytes. However, it is better to present distinct proposals than too cautious ones, the latter of which might fail to be registered in databases. Wrong structures, as they are registered, can be corrected; cautious ones, which will remain unregistered, will also remain anonymous.

## 5. Conclusions

This case study aims to provide insights into which problematics can arise when analyzing auto-MS/MS spectra. They differ from those that are obtained by traditional CID experiments (Figure 1), in which the measurement parameters, specifically the collision energy, are optimized for the analyte. This study demonstrates that auto-MS/MS spectra can be used for identification. However, existing spectral and molecular structure databases are not sufficient alone for the identification process. Additionally, accessing of chemical reference databases is required to (1) compare product ions in MS/MS spectra with literature data; (2) to obtain more information about specific structural features of the compound class; and (3) to find out if the putatively identified metabolite is described in the literature or not. More detailed knowledge about the specialized metabolite analyte profiles in LC–MS/MS analyses can substantially contribute to improved visualization and deeper analysis of compound class patterns by molecular networking [80].

## Figures and Tables

**Figure 1 cells-11-01025-f001:**
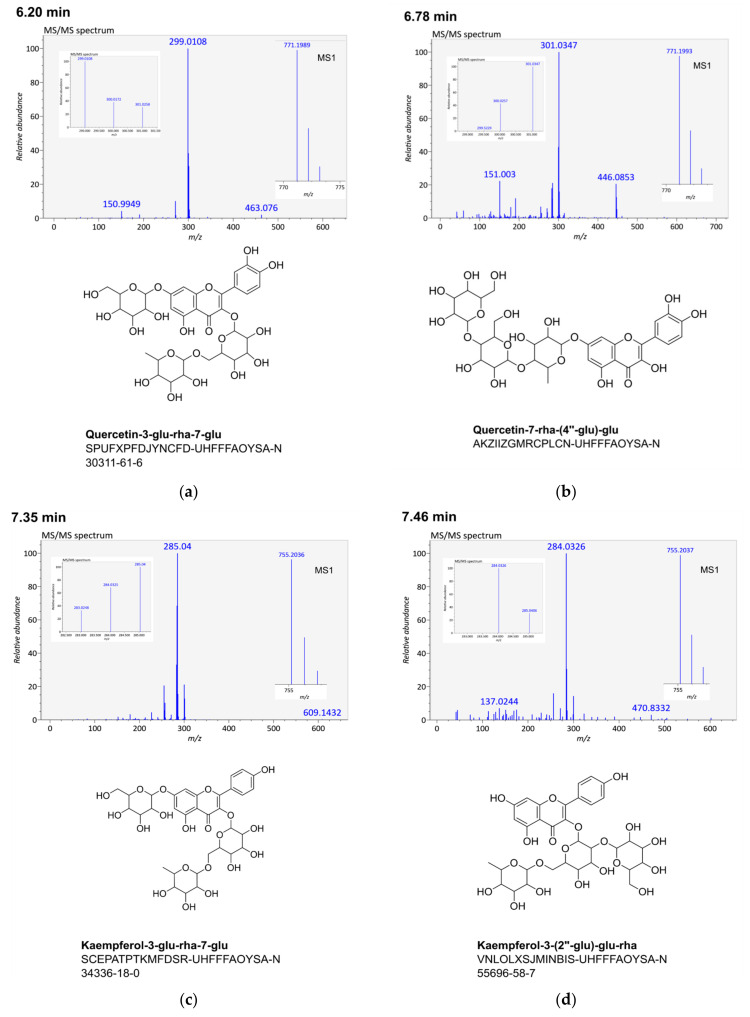
Negative MS/MS (pseudo MS^3^) spectra of 12 selected flavonol glycosides (**a**–**l**), their putative structures, InChI keys, and most closely fitting CAS registry numbers (if available). The illustrated spectra are ordered according to retention times that facilitate their attribution in tables and text.

**Figure 2 cells-11-01025-f002:**
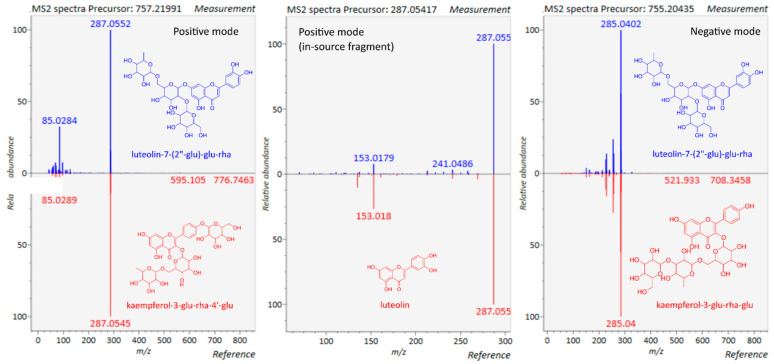
Blue spectra: MS/MS spectra of putative luteolin-7-(2″-glu)-glu-rha in the positive and negative mode and of the in-source fragment ion at 287 Da (luteolin in the positive mode; for experimental condition, see Section 2.6; red spectra: first hit of MS/MS spectral library search; for details, see Section 2.6; software MS-DIAL 4.80 [17].

**Figure 3 cells-11-01025-f003:**
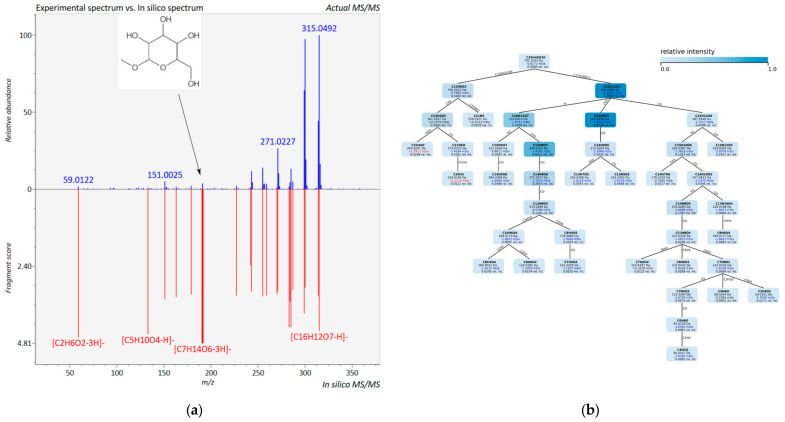
‘*In silico*’ fragmentation of (**a**) MS-Finder [24,73] and (**b**) SIRIUS [25,26,27,28], isorhamnetin-3-(2”-xyl)-glu-7-rha. The figure aims just to provide an idea how the program interface of the two softwares look alike. The analysis data of the 12 flavonol glycosides are available [74,75] and can be viewed by the publicly available software tools.

**Figure 4 cells-11-01025-f004:**
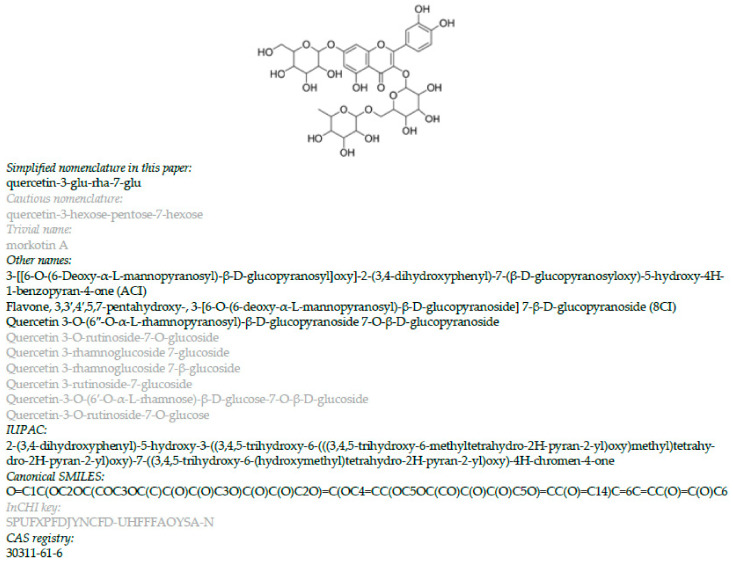
Nomenclatural variants and structure codes for a flavonol glycoside; black: variants more suitable for database registration; grey: variants less suitable for database registration; black bold: IUPAC nomenclature that was required formerly for searching in collective index volumes of Chemical Abstracts or Beilstein.

**Table 1 cells-11-01025-t001:** Positive and negative MS/MS data for 12 selected flavonoids; retention times, putative identification, precursor ion masses, calculated precursor ion masses, Δ ppm values, adduct types, literature/database reference spectra. Abbreviations: glu, glucose; rha, rhamnose; xyl, xylose; Y_0_, flavonoid. Serial letters refer to Figure 1.

Ret.	Putative Structure ^1^	Prec. ion	Calc.	D ppm	Adduct	Lit/Ref.
**6.20**	**Quercetin-3-glu-rha-7-glu**	771.1989	771.1990	−0.1	[M − H]^−^	[46,47,48,49,50]
MS/MS: 463.0761 (<1) M − glu − rha [C_21_H_19_O_12_]^−^, 301.0258 (31) [Y_0_]^−^ [C_15_H_9_O_7_]^−^, 300.0172 (38) [Y_0_ − H]^•−^ [C_15_H_8_O_7_]^•−^, 299.0108 [100) [Y_0_ − 2H]^−^ [C_15_H_7_O_7_]^−^, 271.0152 (10) Y_0_-glu fragm. [41] [C_14_H_8_O_6_]^−^						
**6.78**	**Quercetin-7-rha-glu-glu**	771.1993	771.1990	0.4	[M − H]^−^	
MS/MS: 446.0853 (1) M − glu− glu [C_21_H_18_O_11_]]^•−^, 301.0347 (100) [Y_0_]^−^ [C_15_H_9_O_7_]^−^, 300.0257 (43) [Y_0_ − H]^•−^ [C_15_H_9_O_7_]^•−^, 283.0236 (18) Y_0_-rha fragm. [41] [C_15_H_7_O_6_]^─^, 271.0286 (6) Y_0_-rha fragm. [41] [C_14_H_7_O_6_]^─^, 255.0297 (7) Y_0_-rha fragm. [41] [C_14_H_7_O_5_]^─^						
**7.35**	**Kaempferol-3-glu-rha-7-glu**	755.2030	755.2040	−0.5	[M − H]^−^	[46,49,52,53]
MS/MS: 609.1432 (<1) (M − rha) [C_27_H_29_O_16_]^•−^, 357.0615 (<1) (Y_0_-rha fragm. [41]), [C_18_H_13_O_8_]^─^, 285.0400 (100) [Y_0_]^−^ [C_15_H_7_O_6_]^−^, 284.0325 (68) [Y_0_ ─ H]^•−^ [C_15_H_6_O_6_]^•−^, 283.0284 (33) ] Y_0_ ─ 2H]^−^ [C_15_H_8_O_6_]^−^, 241.0509 (2) (Y_0_-rha fragm. [41]), [C_14_H_9_O_4_]^─^						
	*kaempferol-glu-rha* ^ISF^	593.1518	593.1506	2.0	[M − H]^−^	
	*kaempferol-glu* ^ISF^	447.0934	447.0927	1.6	[M − H]^−^	
**7.46**	**Kaempferol-3-(2″-glu)-glu-rha**	755.2037	755.2040	−0.4	[M − H]^−^	[49,52,54]
	MS/MS: 447.0815 (<1) [C_21_H_19_O_11_]^−^M-glu-rha, 285.04059 [Y_0_]^−^ [C_15_H_9_O_6_]^−^ (31), 284.0326 (100) [Y_0_ ─ H]^•−^ [C_15_H_8_O_6_]^•−^, 256.0348 (16) Y_0_-glu-rha fragm. [C_15_H_12_O_4_]^─^, 241.0523 (3) Y_0_-glu-rha fragm. [C_14_H_9_O_4_]^─^					
**7.67**	**Quercetin-3-glu-rha-7-rha**	757.2192	757.2201	2.0	[M + H]^+^	[55,56,57]
	MS/MS: 303.0497 [C_15_H_11_O_7_]^+^					
		779.2006	779.2005	0.1	[M + Na]^+^	
	MS/MS: 347.0924 (1) [C_12_H_20_O_10_Na]^••+^ (glu-rha), 303.0490 (100) [C_15_H_11_O_7_]^+^; rhaα2glu+Na^+^: 185.0103 (1.1) [C_6_H_10_O_5_Na]^+^, 243.0824 (1.0) [C_9_H_16_O_6_Na]^+^, 331.0998 (13.6) [C_12_H_20_O_9_Na]^+^					
		755.2033	755.2040	−1.0	[M − H]^−^	
	MS/MS: 446.0766 (8) [C_21_H_19_O_11_]^−^ M − glu − rha, 301.0226 (51) [Y_0_]^−^ [C_15_H_9_O_7_]^−^, 300.0252 (79) [Y_0_ ─ H]^•−^ [C_15_H_8_O_7_]^•−^, 299.0108 (100) [Y_0_ ─ 2H]^−^ [C_15_H_7_O_7_]^−^, 271.0126 (24) quercetin-glu fragm. [41] [C_14_H_8_O_6_]^−^, 255.0177 (12) quercetin-glu fragm. [41] [C_14_H_7_O_5_]^─^					
**7.93**	**Quercetin-7-(2″-glu)-glu-rha**	773.2145	773.2096	6.3	[M + H]^+^	
	MS/MS: 303.0501 (100) [Y_0_ + H]^+^ [C_15_H_11_O_7_]^+^					
	*quercetin-glu-rha* ^ISF^	611.1614	611.1607	1.1	[M + H]^+^	
	*quercetin-glu* ^ISF^	465.1031	465.1028	0.6	[M + H]^+^	
	*quercetin* ^ISF^	303.0507	303.0499	2.6	[M + H]^+^	MB: PR309259
	MS/MS: 303.0497 (100) [C_15_H_11_O_7_]^+^, 285.0387 (5) [C_15_H_9_O_6_]^•+^, 257.044 (9) [C_14_H_9_O_5_]^••+^, 229.049 (11) [C_13_H_9_O_4_]^••+^, 165.0179 (11) [C_8_H_5_O_4_]^•••+^, 153.0183 (10) [C_7_H_5_O_4_]^+^, 137.0229 (7) [C_7_H_5_O_3_]^•+^					
		795.1963	795.1954	1.1	[M + Na]^+^	
	MS/MS: 493.1517 (4) [C_18_H_30_O_14_Na]^••+^ glu-rha-glu − OH, 347.0924 (1) [C_12_H_20_O_10_Na]^••+^ glu-rha, 303.0490 (100) [Y_0_ + H]^+^ [C_15_H_11_O_7_]^+^; rhaα6glu+Na^+^: 185.0410 (1.6) [C_6_H_10_O_5_Na]^+^, 331.0954 (5.5) [C_12_H_20_O_9_Na]^+^ ; gluβ2glu+Na^+^: 185.0410 (1.6) [C_6_H_10_O_5_Na]^+^, 245.0499 (0.8) [C_9_H_14_O_7_Na]^+^, 259.0483 (0.5) [C_9_H_16_O_7_Na]^+^, 329.0226 (1.0) [C_12_H_18_O_9_Na]^+^, 347.0983 (1.0) [C_12_H_20_O_10_Na]^+^					
		771.1990	771.1989	0.1	[M − H]^−^	
	MS/MS: 301.0340 (100) [C_15_H_9_O_7_]^−^					
		1543.4070	1543.4045	1.6	[2M − H]^−^	
	MS/MS: 755.3816 (<1) ≈ [C_33_H_40_O_20_]^•−^, 301.0336 (57) [Y_0_]^−^ [C_15_H_9_O_7_]^−^, 300.0266 (100) [Y_0_ ─ H]^•−^ [C_15_H_8_O_7_]^•−^, 271.0243 (59) quercetin-glu fragm. [41] [C_14_H_8_O_6_]^−^, 255.0294 (35) quercetin-glu fragm. [41] [C_14_H_7_O_5_]^─^					
**8.04**	**Quercetin-3-(2″-rha)-glu-rha**	755.2015	755.2040	−3.3	[M − H]^−^	[56,58,59,60]
	MS/MS: 489.0837 (<1) [C_23_H_21_O_12_]^••−^ (M − rha − rha), 301.0343 (35) [Y_0_]^−^ [C_15_H_9_O_7_]^−^, 300.0275 [Y_0_ ─ H]^•−^ (100) [C_15_H_8_O_7_]^•−^, 299.0198 (14) [Y_0_ ─ 2H]^−^ [C_15_H_7_O_7_]^−^, 271.0248 (25) quercetin-glu fragm. [41] [C_14_H_8_O_6_]^−^, 255.0301 (12) quercetin-glu fragm. [C_14_H_7_O_5_]^─^ [41]					
**8.56**	**Luteolin-7-(2″-glu)-glu-rha**	757.2203	757.2186	2.2	[M + H]^+^	
	MS/MS: 287.0551 (100) [C_15_H_11_O_6_]^+^					
	*luteolin* ^ISF^	287.0550	287.0551	−0.3	[M + H]^+^	BMDMS-NP 29525
	MS/MS: 287.055 (100) [C_15_H_11_O_6_]^+^, 241.0486 (3) [C_14_H_9_O_4_]^••+^, 213.0547 (3) [C_13_H_9_O_3_]^••+^, 153.0179 (8) [C_7_H_5_O_4_]^+^					
		779.2012	779.2005	0.7	[M + Na]^+^	
	MS/MS: 347.0893 (0.2) glu-rha [C_12_H_20_O_10_Na]^●●+^, 287.0538 (100)) [C_15_H_11_O_6_]^+^, 203.0529 (5) glucose [C_6_H_12_O_6_Na]^+^; rhaα6glu+ Na^+^: 185.0080 (1.4) [C_6_H_10_O_5_Na]^+^, 243.2668 (0.4) [C_9_H_16_O_6_Na]^+^, 331.0982 (26.2) [C_12_H_20_O_9_Na]^+^ ; gluβ2glu+Na^+^: 185.0080 (1.4) [C_6_H_10_O_5_Na]^+^, 245.0467 (1.1) [C_9_H_14_O_7_Na]^+^, 329.0215 (4.9) [C_12_H_18_O_9_Na]^+^, 347.0983 (2.4) [C_12_H_20_O_10_Na]^+^					
		755.2039	75.2040	−0.1	[M − H]^−^	
	MS/MS: 327.0515 (38) [C_17_H_11_O_7_]^••−^, 285.0400 (100) [Y_0_]^−^ [C_15_H_10_O_6_]^−^, 284.0325 [Y_0_ ─ H]^•−^ (53) [C_15_H_9_O_6_]^•−^					
**8.70**	**Isorhamnetin-3-(2″-glu-6″-feruloyl)-glu**	815.2037	815.2040	−0.4	[M − H]^−^	
	MS/MS: 451.1647 (1) ferulic acid-glu + sugar fragm. [C_21_H_29_O_11_]^─^ (MS-Finder), 357.0563 (2) isorhamnetin-glu fragm. [C_15_H_17_O_10_]^─^, 315.0490 (47) [Y_0_ ]^−^ [C_16_H_11_O_7_]^−^, 314.0423 (100) [Y_0_ ─ H]^•−^ [C_16_H_10_O_7_]^•−^, 301.0285 (20) [Y_0_ ─ CH_2_]^−^ [C_15_H_9_O_7_]^−^, 300.0251 [Y_0_–CH_2_–H]^•−^ (100) [C_15_H_8_O_7_]^•−^, 175.0395 (8) [C_10_H_7_O_3_]^••−^ ferulic acid					
**8.72**	**Isorhamnetin-3-(2″-xyl)-glu-7-rha**	757.2193	757.2186	0.9	[M + H]^+^	[55]
	MS/MS: 287.0551 (100) [C_15_H_11_O_6_]^+^					
		779.2010	779.2005	0.6	[M + Na]^+^	
		755.2045	755.2040	0.7	[M − H]^−^	
	MS/MS: 357.0594 (<1) isorhamnetin-glu fragm. [C_15_H_17_O_10_]^─^, 315.0492 (100) [Y_0_]^−^ [C_16_H_11_O_7_]^−^, 314.0413 (44) [Y_0_ ─ H]^•−^ [C_16_H_10_O_7_]^•−^, 301.0297 (19) [Y_0_ ─ CH_2_]^−^ [C_15_H_9_O_7_]^−^, 300.0258 (97) [Y_0_–CH2–H]^•−^ [C_15_H_9_O_7_]^•−^, 299.0183 (64) [Y_0_–CH_2_–2H]^−^ [C_15_H_7_O_7_]^−^					
**9.11**	**Quercetin-3-(3″-*p*-coumaroyl)-glu-rha**	755.1826	755.1829	−0.4	[M − H]^−^	[59]
	MS/MS: 423.0176^2^ (1) [C_21_H_11_O_10_]^8^^●−^, 395.0331^2^ (10) [C_20_H_11_O_9_)^6^^●−^, (querc. fragm. + *p*-coum.a.), 301.032 (25) [C_15_H_9_O_7_]^−^, 300.0259 (100) [C_15_H_9_O_7_]^•−^, 271.0240 (33) Y_0_-glu-rha fragm. [41] [C_14_H_8_O_6_]^−^, 255.00303 (19), 147.045 (1) [C_9_H_7_O_2_]^−^ (*p*-coumaric a.)					
	*quercetin-3-(3’’-p-coumaric a.)-glu^•^* ^ISF^	609.1249^2^	609.1250	34.0	[M − H]^−^	
**9.50**	**Isorhamnetin-3-(6″-feruloyl)-glu-7-glu**	815.2040	815.2040	0	[M − H]^−^	
	MS/MS: 315.0467 (18) [C_16_H_11_O_7_]^−^, 314.0421 (77) [C_16_H_10_O_7_]^•−^, 300 (26) [C_15_H_9_O_7_]^•−^, 299.019 (100) [C_15_H_7_O_7_]^−^, 175.030 (1) [C_10_H_7_O_3_]^••−^ (ferulic a.)					

^ISF^, in-source fragmentation ions (italics); ^1^ MS/MS data do not allow differentiation of glucose and galactose, and xylose, arabinose, or apiose; ^2^ for possible feature structure, see Appendix A.

**Table 2 cells-11-01025-t002:** MS-Dial 4.80 aligment table excerpt (ID, alignment ID; RT, retention tine in min; *m*/*z*, product ion; type, product ion type; metabolite, identification or n.i., not identified; S/N, signal–noise ratio; *p*, *p* value calc. on basis of an ANOVA; fold change (min–max); bar chart of signal intensities in different sample groups, left: blank, samples *Ranunculus auricomus* flower buds; red colors: diploids; green colors, tetraploids, pink/violett colors, hexaploids; first group of three bars, 10 h photoperiod; second group of three bars, 16 h photoperiod (from the left; for additional experimental details, see Ref. [36]). Related analytes show similar bar chart patterns and are highlighted in red. Luteolin-7-(2″-glu)-glu-rha is detected as [M + H]^+^ and [M + Na]^+^ adduct. In addition, an in-source fragment of luteolin is formed.

ID	RT	*m*/*z*	Type	Metabolite	S/N	*p* (ANOVA)	Fold Change	Bar Chart
12155	8.56	757.2199	[M + H]^+^	luteolin-7-(2″-glu)-glu-rha	380.1	7.61 × 10^−7^	338.04	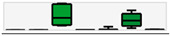
12407	8.56	795.1692	[M + K]^+^	n.i.	21.0	1.84 × 10^−7^	2536.07	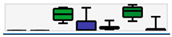
3569	8.56	287.0542	[M + H]^+^	luteolin	27.5	3.10 × 10^−6^	39.82	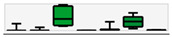
12308	8.56	779.2012	[M + Na]^+^	luteolin-7-(2”-glu)-glu-rha	261.4	5.80 × 10^−11^	301.71	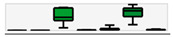
4544	8.56	328.1361	[M + H]^+^	n.i.	6.5	4.71 × 10^−1^	3.57	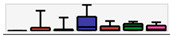
3599	8.56	288.1437	[M + H]^+^	n.i.	4.6	7.19 × 10^−1^	2.15	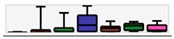

**Table 3 cells-11-01025-t003:** Molecular structure database analysis results comparison of MS-FINDER (HMDB, LipidMAPS, FoodDB, PlantCyc, ChEBI, Natural Product Atlas, NANPDB, COCONUT, KNApSAck, PubChem, UNPD) and SIRIUS (BioCyc, ChEBI, COCONUT, GNPS, HMDB, KEGG, KNApSAck, Natural Products, Plantcyc, PubChem) for 12 flavonol glycoside negative MS/MS spectra. In only one case, the first hit of the structure base search corresponded to the proposed structure (marked in red).

Flavonol Glycoside	MS_FINDER 1st	SIRIUS 1st	Identification (Rank Number of Proposed Structure)
Que-3-glu-rha-7-glu	Que-3-glu-rha-7-gal		Que-3-glu-rha-7-glu (4 ^1^)
Que-7-rha-glu-glu	Kae-3-(2″-glu)-glu-rha		
Kae-3-glu-rha-7-glu	Kae-3-glu-glu-7-rha	Kae3-glu-rha-glu	Kae-3-glu-rha-7-glu (10) ^1,2^
** Kae-3-(2″-glu)-glu-rha **	Kae-3-glu-glu-7-rha	** Kae-3-(2″-glu)-glu-rha **	Kae-3-(2″-glu)-glu-rha (8 ^1^, 1 ^2^)
Que-3-glu-rha -7-rha	Kae-3-glu-glu-7-rha		Que-3-glu-rha -7-rha (6 ^1^)
Que-7-(2″-glu)-glu-rha	Kae-3-glu-glu-glu	Que-3-glu-rha-glu	Que-7-(2″-glu)-glu-rha (5 ^1^)
Que-3-(2″-rha)-glu-rha	Kae-3-glu-glu-7-rha	Que-3-glu-rha-rha	Que-3-(2″-rha)-glu-rha (2 ^1^)
Lut-7-(2″-glu)-glu-rha	Kae-3-glu-glu-7-rha	Kae-3-glu-rha-glu	
Iso-3-(2″-glu)-glu-ferul.			
Iso-3-(2″-xyl)-glu-7-rha	Kae-3-glu-glu-7-rha	Kae-3-glu-glu-rha	Iso-3-(2″-xyl)-glu-7-rha (13 ^1^,45 ^2^)
Que-3-(3″-coum)-glu-rha	Que-3-rha-glu-coum	Kae-3-glu-rha-sinap	Que-3-(3″-coum)-glu-rha (17 ^1^)
Iso-3-glu-ferul-7-glu	Kae-3-(2″-coum-glu)-glu	Que-3-(2″-sinap)-glu-rha	

^1^ MS-Finder; ^2^ SIRIUS; Iso, Isorhamnetin; Kae, Kaempferol; Que, Quercetin; gal, galactose; glu, glucose; rha, rhamnose, xyl, xylose; coum, p-coumaric acid; ferul, ferulic acid; sinap, sinapic acid.

## Data Availability

Raw and analysis data are available at Goettingen Research Online. They include spectral data in MassBank [81], NIST [82], and Mascot format [83]. Additionally, for all 12 flavonoid glycosides, the project folders of the MS-FINDER [74] and SIRIUS [75] analyses are provided.

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
