# Peer review of "Secondary or Specialized Metabolites, or Natural Products: A Case Study of Untargeted LC–QTOF Auto-MS/MS Analysis"

_cells, 2022, doi:10.3390/cells11061025_

Round 1

Reviewer 1 Report

Dear editor,

Thank you for inviting me to evaluate the article titled “Secondary or Specialized Metabolites, or Natural Products: a Case Study of Untargeted LC–QTOF Auto‐MS/MS Analysis”. I have carefully read the manuscript, the Introduction gave a satisfactory literature survey on the similar topic and it outlined the proposed method well. Appropriate figures and tables were given to make the paper understood easily. Besides, there are no experimental ethics, conflict of interest, plagiarism or publication ethics problems in this MS. However, there are some errors that must be corrected and some questions that must be clarified. 

The comments are as follows:

(1) Page 12 and Page 13, the author illustrated full collision energy ramp‐MS2 spectra were proved as helpful to structure identification. However, some compounds own rigid structures and can’t formed fragment ions in CID. The author should explain the methods for identifying this compounds in the article.

(2) Page 13, the author demonstrated in‐source fragmentation is recognized as a useful phenomenon in auto‐MS/MS spectral analysis. Although in-source fragmentation is beneficial for structure identification, it can interfere the discrimination of metabolites in the absence of chromatographic separation. So the author may emphasize the importance of chromatographic separation by utilizing in‐source fragmentation.

(3) Line 17, “do not only” should be corrected to “not only”.

(4) Line 34-36 and Line 48-51,the expression of the sentence is too complicated, please concise it and split it into two sentences.

(5) Lin 36-37, “access to” should be corrected to “accessing”.

(6) Line 63-66, There are grammatical errors in this sentence.

(7) Line 80, “ elucidation—identification ” should be corrected to “elucidation and identification”.

(8) Figure 1 is a combination of several figures. Please add the serial numbers of the figures.

(9) Please change Figure 2 (b) to a table. Barchart can be used as a separate figure.

(10) Lin 459, “aims at providing” should be corrected to “aim to provide”.

Author Response

Dear Reviewer 1,

Thank you for your comments!

(1) Page 12 and Page 13, the author illustrated full collision energy ramp‐MS2 spectra were proved as helpful to structure identification. However, some compounds own rigid structures and can’t formed fragment ions in CID. The author should explain the methods for identifying this compounds in the article.

The obtained MS/MS do not correspond to CID spectra that are obtained by optimized collision energy but to auto-MS/MS spectra. The sequential cleavage of for example sugar moieties, as it can be observed in CID studies in an MS2, MS3, MS4, … spectrums series. In addition, the higher fragmentor voltage aims at obtaining more in-source fragmentation. Their MS/MS spectra are suggested to replace the spectral information of a CID experiment, which works in some cases and not in many other cases. In-source fragmentation does not happen as a rule. Sometimes the whole analyte becomes fragmented is such a way that the spectrum resembles an EI-spectrum.

(2) Page 13, the author demonstrated in‐source fragmentation is recognized as a useful phenomenon in auto‐MS/MS spectral analysis. Although in-source fragmentation is beneficial for structure identification, it can interfere the discrimination of metabolites in the absence of chromatographic separation. So the author may emphasize the importance of chromatographic separation by utilizing in‐source fragmentation.

I agree. One way to find out of this dilemma is to use the intensity pattern of an analyte in different sample groups (former Figure 2b, now Table 2). There may be solutions with recently developed software tools to clean features but in that case, you never know if all the important information is retained by the cleaning process.

I also agree to the importance of longer elution gradients. I have pointed this out in section 4.2 already in the original version and I extended the text now to emphasize that every kind of spectrometric detection benefits from better chromatographic separations.

(3) Line 17, “do not only” should be corrected to “not only”.

Corrected.

(4) Line 34-36 and Line 48-51, the expression of the sentence is too complicated, please concise it and split it into two sentences.

Corrected.

(5) Lin 36-37, “access to” should be corrected to “accessing”.

Corrected.

(6) Line 63-66, There are grammatical errors in this sentence.

I rephrased the sentence, hopefully with less grammatical errors.

(7) Line 80, “ elucidation—identification ” should be corrected to “elucidation and identification”.

Corrected. I split the sentence into two in attempts to avoid a too long one.

(8) Figure 1 is a combination of several figures. Please add the serial numbers of the figures.

Serial letters have been added (according to the journal guidelines).

(9) Please change Figure 2 (b) to a table. Barchart can be used as a separate figure.

Figure 2b has been changed to a table. However, I kept the bar charts integrated because they provide a substantial portion of information for the table that would be lost if shown separately.

(10) Lin 459, “aims at providing” should be corrected to “aim to provide”.

Corrected.

I have rechecked the whole text for typos, readability and conciseness. I uploaded a separate version of the corrected manuscript with tracked changes.

Reviewer 2 Report

-In this manuscript, the authors researched the “Secondary or Specialized Metabolites, or Natural Products: a Case Study of Untargeted LC–QTOF Auto‐MS/MS Analysis”. This paper can be interesting in MS/MS Analysis domain. They discussed about specialized metabolites of plant tissues.  I affirm its acceptance for publication. There are some concerns before it is suggestion.

Significance of metabolic activity (metabolomics) in various cells has been recently documented. This article more deeply discussed about metabolomics subsets, and analytical metabolic difference. I encourage to cite this (https://doi.org/10.3390/ijms22031160 and https://doi.org/10.1016/j.procbio.2020.08.023). Author should list about specialized figure 1 metabolites in conclusion part.

Author Response

Dear Reviewer,

Thank you for your comments!

-In this manuscript, the authors researched the “Secondary or Specialized Metabolites, or Natural Products: a Case Study of Untargeted LC–QTOF Auto‐MS/MS Analysis”. This paper can be interesting in MS/MS Analysis domain. They discussed about specialized metabolites of plant tissues.  I affirm its acceptance for publication. There are some concerns before it is suggestion.

Significance of metabolic activity (metabolomics) in various cells has been recently documented. This article more deeply discussed about metabolomics subsets, and analytical metabolic difference. I encourage to cite this (https://doi.org/10.3390/ijms22031160 and https://doi.org/10.1016/j.procbio.2020.08.023) . Author should list about specialized figure 1 metabolites in conclusion part.

The first article that you pointed out to me is

Raja, G.; Gupta, H.; Gebru, Y.A.; Youn, G.S.; Choi, Y.R.; Kim, H.S.; Yoon, S.J.; Kim, D.J.; Kim, T.-J.; Suk, K.T.Recent Advances of Microbiome-Associated Metabolomics Profiling in Liver Disease: Principles, Mechanisms, and Applications. IJMS 2021, 22, 1160, doi:10.3390/ijms22031160.

It represents an interesting review about metabolomic analyses in context with liver diseases. The review focuses on central metabolites and secondary bile acids but not on specialized or secondary metabolites as specifically mentioned in the title. Unfortunately, I see no way how to include it into the manuscript.

The second article that you recommended to include is

Raja, G.; Jung, Y.; Jung, S.H.; Kim, T.-J. 1H-NMR-based metabolomics for cancer targeting and metabolic engineering –A review. Process Biochemistry 2020, 99, 112–122, doi:10.1016/j.procbio.2020.08.023.

It deals with proton NMR of central metabolites. Similarly as with the first paper, I see no way to cite it in the manuscript. Again, the focus should have been set on specialized or secondary metabolites and the method should be LC–NMR, in which the NMR functions as alternative detector in the chromatography.

I referred to Figure 1 in the conclusion.

I have rechecked the whole text for typos, readability and conciseness. I uploaded a separate version of the corrected manuscript with tracked changes.

Reviewer 3 Report

Manuscript title: Secondary or Specialized Metabolites, or Natural Products: a Case Study of Untargeted LC–QTOF Auto‐MS/MS Analysis  

General comment: In this paper, the author discuss about the possibilities and limitations of the identification of twelve partially flavonol glycosides, characteristic specialized metabolites of plant tissues, some of them isomeric and isobaric. I find it suitable for publication on Cells after major revisions. Several points should be issued. The main ones are:

  • L85: “Spectral databases are incomplete, as the mostly are limited to spectra from commercially available standards. In attempts to overcome this constraint, molecular structure databases have been recommended as primary search tool for unknowns [23].” This section should be upgraded. For example, several databases ad hoc created for the identification of specific classes of compounds exist. See: Profiling of quercetin glycosides and acyl glycosides in sun-dried peperoni di Senise peppers (Capsicum annuum L.) by a combination of LC-ESI (-)-MS/MS and polarity prediction in reversed-phase separations. Analytical and Bioanalytical Chemistry, 412(12), 3005-3015.
  • L122: Which vegetable specie? Please, provide complete information.
  • Section 2.2. Please, report the weight of the lyophilized buds extracted with 1 mL of MeOH.
  • L132: Were the samples filtered before the LC-MS analysis?
  • Section 2.6: The authors should report the fragmentation nomenclatures used for aglycones and glycoconjugates.
  • L193: “thus a hexose was assumed to be a glucose, a deoxyhexose a rhamnose and a pentose a xylose.” This assumption is not properly correct, as many glycosylated flavonoids contain also arabinose or galactose as saccharide units. An unambiguous identification in this sense is only possible with the NMR, thus the authors should report the name of the substituents as Glu/gal, ara/rha etc, or generically pentose, hexose.
  • Section 3. I suggest the author to better organize the contents. The strategy used for the identification should be discussed as first, and then the identified compounds should be reported. Moreover, I suggest to organize better the Figure 1, for example also reporting the fragmentation pathway of the new flavonoids, or the structures of the MSMS fragments. Morover, an explanation of the inserts should be added in the relative captions.
  • Figure 3 b. Please zoom in the figure.

Author Response

Dear Reviewer 3,

Thank you for your comments! 

  • L85: “Spectral databases are incomplete, as the mostly are limited to spectra from commercially available standards. In attempts to overcome this constraint, molecular structure databases have been recommended as primary search tool for unknowns [23].” This section should be upgraded. For example, several databases ad hoc created for the identification of specific classes of compounds exist. See: Profiling of quercetin glycosides and acyl glycosides in sun-dried peperoni di Senise peppers (Capsicum annuum L.) by a combination of LC-ESI (-)-MS/MS and polarity prediction in reversed-phase separations. Analytical and Bioanalytical Chemistry, 412(12), 3005-3015.

Thank you for pointing out this paper to me. It is nice study with CID experiments. In contrast to that study, auto MS/MS spectra were available in this study. This technology does not allow to use optimized collision energies for each analyte, as it is possible in CID studies. Therefore, the spectra look different. The number of fragment ions from a specific precursor are higher. In attempts to obtain something comparable to CID spectra, higher fragmentor voltages are used that increase in-source fragmentation. In optimal cases, the resulting product ions yield further pseudo-MS/MS spectra.

I could not access the therein-mentioned QUEdb database. The supplementary tables S1-S12 are missing on the journal’s website.

The sentence that you pointed out was meant in a more general sense. If you use software tools, such as MS-FINDER or SIRIUS, a set of structure databases is available which, together with in-silico generated spectra, provide assistance to identify the compound or subcompound class of an unknown. CID spectral databases may be highly helpful if you have CID spectra, but they are only of limited help in case of auto-MS/MS spectra because of the stronger fragmentation, which resembles more an EI- instead of a CI-MS spectrum.

  • L122: Which vegetable specie? Please, provide complete information.

Ranunculus auricomus is not a vegetable (lines 102–106). The investigated buds contain about 40–50 flavonoids alone and the presented data only comprise a portion that was selected for illustrative purposes. The complete research will be published as soon as the identification process is finished.

  • Section 2.2. Please, report the weight of the lyophilized buds extracted with 1 mL of MeOH.

    These and other experimental details will be published in separate paper that will contain the complete metabolite analysis.

  • L132: Were the samples filtered before the LC-MS analysis?

No. All of them were diluted to 0.1 mg/mL to avoid signal concentrations outside of the linear range.

  • Section 2.6: The authors should report the fragmentation nomenclatures used for aglycones and glycoconjugates.

2.6 has been extended. I added semoisynthetic IUPAC names into table S1.

  • L193: “thus a hexose was assumed to be a glucose, a deoxyhexose a rhamnose and a pentose a xylose.” This assumption is not properly correct, as many glycosylated flavonoids contain also arabinose or galactose as saccharide units. An unambiguous identification in this sense is only possible with the NMR, thus the authors should report the name of the substituents as Glu/gal, ara/rha etc, or generically pentose, hexose.

I agree to everything that you write except that is possible to differentiate between glucose and galactose when you have a [M+Na]+ MS/MS spectrum (ref. 39). but if you stick to the certainly justified ambiguous nomenclature of hexoses, deoxyhexoses and pentoses, you risk that the structure is not indexed in Chemicals Abstracts. This happened also to the above-mentioned paper about the pepper flavonol glycosides. Nine out of 15 flavonoid glycoside structures were not incorporated into the index due to the ambiguous nomenclature. Therefore, if somebody comes up with a similar structure during the identification process and performs a structure search in SciFinder, the paper will be missed. Actually, I am not aware of any more comprehensive database and it probably also contains the most comprehensive collection of studies that applied NMR for structure analysis. Another possibility would be Reaxys but German universities have set a ban on Elsevier products.

Section 3. I suggest the author to better organize the contents. The strategy used for the identification should be discussed as first, and then the identified compounds should be reported.

I decided to separate results and discussion into two section. I admit that my result section also contains text portions that resemble a combined result and discussion section in many natural product reports. My intention was to separate the identification part for the twelve flavonol glycosides from a rather more general discussion of various aspects of the identification process.

I have extended the first paragraph to outline the identification strategy. However, it is also described in Materials and Methods.

Moreover, I suggest to organize better the Figure 1, for example also reporting the fragmentation pathway of the new flavonoids, or the structures of the MSMS fragments.

Figure 1 focuses on the negative ionization spectra of the twelve flavonol glycosides. The auto-MS/MS setup generates spectra that are different from CID experiments and I have repeatedly pointed out the differences in the text. Figure 1 also aims to provide the reader with a general overview how the spectra look like. Actually, they are pseudo-MS/MS spectra. In case of the flavonol glycosides, the harsher fragmentation conditions generate spectra in which the flavonoid ring system is more prominent and you obtain more information about the sugar substitution patterns than about sugar structures. Figure 1 specifically focuses on this information.

The auto-MS/MS spectra do not contain as much information about the glycoside structure as CID experiments would offer. For this reason, data from a total fragmentation study of flavonol glycosides, detectable in source fragmentation and their MS/MS spectra (if available), and from CID studies (as far as possible) were used in combination to develop structure proposals. These were checked in the literature for feasibility. The whole process is described in a separate result section for each flavonol glycoside.

Table 1 contains all the information, including elucidation of fragmentation pathways (as far as possible) such as you find it in many similar studies.

Sometimes fragment structures are difficult to figure out because many isomers can exist. For quercetin-3-(3''-p-coumaroyl)-glu-rha Figure S1 offers a structure proposal for some fragment ions that offer specific structural information.  

Morover, an explanation of the inserts should be added in the relative captions.

Unfortunately, I do not understand what you mean. Could you please point out a specific example to me?

  • Figure 3 b. Please zoom in the figure.

The main purpose of Figure 3 is to illustrate the program interface of the two software tools. For those who interested in more details, the compelte set of the analyses results are available at the specified repository. Both software tools are available freely.

I have rechecked the whole text for typos, readability and conciseness. I uploaded a separate version of the corrected manuscript with tracked changes.

Round 2

Reviewer 3 Report

General comment: Most of the comments and suggestions raised have been properly resolved by the author. I find the paper suitable for publication on Cells after minor revisions

  • “It is nice study with CID experiments. In contrast to that study, auto MS/MS spectra were available in this study. This technology does not allow to use optimized collision energies for each analyte, as it is possible in CID studies. Therefore, the spectra look different. The number of fragment ions from a specific precursor are higher. In attempts to obtain something comparable to CID spectra, higher fragmentor voltages are used that increase in-source fragmentation. In optimal cases, the resulting product ions yield further pseudo-MS/MS spectra.”

The answer of the author is very exhaustive. I suggest clarifying this point also in the main text of the manuscript.

  • “I could not access the therein-mentioned QUEdb database. The supplementary tables S1-S12 are missing on the journal’s website.”

I think it is useful for the author to refer to the following database (available as “Electronic supplementary material” at the link: https://link.springer.com/article/10.1007/s00216-020-02547-2#Sec12) also to support the identification of his own compounds, since many of them are quercetin derivatives.

  • Section 2.2. These and other experimental details will be published in separate paper that will contain the complete metabolite analysis.

Even though other experimental details will be published in another paper, the information about the experimental details of this work should be completed. 

Author Response

Dear Reviewer 3,

Thank you for your comments. 

  • “It is nice study with CID experiments. In contrast to that study, auto MS/MS spectra were available in this study. This technology does not allow to use optimized collision energies for each analyte, as it is possible in CID studies. Therefore, the spectra look different. The number of fragment ions from a specific precursor are higher. In attempts to obtain something comparable to CID spectra, higher fragmentor voltages are used that increase in-source fragmentation. In optimal cases, the resulting product ions yield further pseudo-MS/MS spectra.”

The answer of the author is very exhaustive. I suggest clarifying this point also in the main text of the manuscript.

The above-mentioned statements of mine can be found in the text of the manuscript, lines 190–198 in the Results, and 358–376 in the Discussion. Most of them were already present in the first sumitted version.

  • “I could not access the therein-mentioned QUEdb database. The supplementary tables S1-S12 are missing on the journal’s website.”

I think it is useful for the author to refer to the following database (available as “Electronic supplementary material” at the link: https://link.springer.com/article/10.1007/s00216-020-02547-2#Sec12 ) also to support the identification of his own compounds, since many of them are quercetin derivatives.

I included the suggested reference in line 189 [45].

  • Section 2.2. These and other experimental details will be published in separate paper that will contain the complete metabolite analysis.

 Even though other experimental details will be published in another paper, the information about the experimental details of this work should be completed. 

I added information about extract yield to Material and Methods (lines 130–131). There was quite some variation in the obtained amounts, which did not correlate with extracted plant tissue amounts. These differences were probably caused by differently advanced development of the flower petals in the buds. These circumstances will be paid attention in their statistical evaluation, which is not part of this manuscript. This manuscript’s focus is on structure elucidation alone.

Some minor text and format corrections have been added to the uploaded manuscript version.